# Phenomena of Intussusceptive Angiogenesis and Intussusceptive Lymphangiogenesis in Blood and Lymphatic Vessel Tumors

**DOI:** 10.3390/biomedicines12020258

**Published:** 2024-01-23

**Authors:** Lucio Díaz-Flores, Ricardo Gutiérrez, Miriam González-Gómez, Maria del Pino García, Jose-Luis Carrasco-Juan, Pablo Martín-Vasallo, Juan Francisco Madrid, Lucio Díaz-Flores

**Affiliations:** 1Department of Basic Medical Sciences, Faculty of Medicine, University of La Laguna, 38071 Tenerife, Spainjcarraju@ull.edu.es (J.-L.C.-J.);; 2Instituto de Tecnologías Biomédicas de Canarias, University of La Laguna, 38071 Tenerife, Spain; 3Department of Pathology, Eurofins Megalab-Hospiten Hospitals, 38100 Tenerife, Spain; mpgarcias@megalab.es; 4Department of Bioquímica, Microbiología, Biología Celular y Genética, University of La Laguna, 38206 Tenerife, Spain; pmartin@ull.edu.es; 5Department of Cell Biology and Histology, School of Medicine, Campus of International Excellence “Campus Mare Nostrum”, IMIB-Arrixaca, University of Murcia, 30100 Murcia, Spain; jfmadrid@um.es

**Keywords:** intussusceptive angiogenesis, intussusceptive lymphangiogenesis, benign vessel tumor, malignant vessel tumors, intravascular intussusceptive structures

## Abstract

Intussusceptive angiogenesis (IA) and intussusceptive lymphangiogenesis (IL) play a key role in the growth and morphogenesis of vessels. However, there are very few studies in this regard in vessel tumors (VTs). Our objective is to assess the presence, characteristics, and possible mechanisms of the formation of intussusceptive structures in a broad spectrum of VTs. For this purpose, examples of benign and malignant blood and lymphatic VTs were studied via conventional procedures, semithin sections, and immunochemistry and immunofluorescence microscopy. The results demonstrated intussusceptive structures (pillars, meshes, and folds) in benign (lobular capillary hemangioma or pyogenic granuloma, intravascular papillary endothelial hyperplasia or Masson tumor, sinusoidal hemangioma, cavernous hemangioma, glomeruloid hemangioma, angiolipoma, and lymphangiomas), low-grade malignancy (retiform hemangioendothelioma and Dabska tumor), and malignant (angiosarcoma and Kaposi sarcoma) VTs. Intussusceptive structures showed an endothelial cover and a core formed of connective tissue components and presented findings suggesting an origin through vessel loops, endothelialized thrombus, interendothelial bridges, and/or splitting and fusion, and conditioned VT morphology. In conclusion, the findings support the participation of IA and IL, in association with sprouting angiogenesis, in VTs, and therefore in their growth and morphogenesis, which is of pathophysiological interest and lays the groundwork for in-depth molecular studies with therapeutic purposes.

## 1. Introduction

Intussusceptive angiogenesis and intussusceptive lymphangiogenesis are physio-pathological processes by which preexisting blood and lymphatic vessels divide and expand (microvascular growth), arborize (intussusceptive arborization), remodel (intussusceptive branching remodeling), and segment (intussusceptive segmentation) [1,2,3,4,5,6,7,8,9,10,11,12,13,14,15,16]. These physio-pathological processes of vessel splitting are complementary to sprouting angiogenesis and have synergistic interaction [17,18]. Intussusceptive phenomena are evidenced by the presence in the vascular lumen of newly formed structures (hereinafter intussusceptive structures), which show a cover formed of endothelial cells and a core formed of connective tissue components. The definition and size of the intussusceptive structures that participate in the above-mentioned processes have presented difficulties in comparing results from different studies [19]. Added to this is the different terminology used for them. The best-known intussusceptive structures are the so-called pillars, posts, or transluminal columns, the hallmark of intussusceptive angiogenesis, which facilitate vessel growth through vessel division (intussusceptive microvascular growth). Pillars show a cylindrical morphology and have a diameter less than or equal to 2.5 μm, although some authors extend it up to 4μm [19,20]. Pillars are also called holes because of their appearance when vascular corrosion casting is used. The definition and terminology used for intussusceptive structures of dimensions larger than 2.5/4 μm is even more complex. These bigger structures have been called interstitial tissue structures, meshes, tissue islands, papillae, and tuffs, and large or giant pillars when they become transluminal. They have also been called hollows (image in vascular corrosion casting). Finally, folds have been used for when the intussusceptive structures are not transluminal. In the present work, we will use the terms pillars (≤4 μm) and meshes (≥4 μm) for the intussusceptive structures that are the morphological expression of intussusceptive angiogenesis, encompassing folds in these terms.

Intussusceptive angiogenesis has been described in many physio-pathological conditions, including tumors, such as adenocarcinoma xenografts [5], mammary gland tumors [21], non-Hodgkin lymphomas [22], primary metastatic melanomas [23,24], and gliomas [25,26]. However, this is not the case in vessel tumors and malformative or reactive vessel pseudo-tumors (hereinafter vessel tumors and pseudo-tumors: vessel tumors), in which intussusceptive angiogenesis has been considered only in some benign types [27,28,29,30] and experimentally in multicavernous malformations [31]. The study of intussusceptive phenomena, including a broader spectrum of benign and malignant tumors of blood and lymphatic vessels, is not only of interest to explore the participation of intussusception in them, but also to evaluate the morphological findings that suggest how pillars and meshes are formed in these entities, as well as whether the presentation of these intussusceptive structures is a common fact for all vessel tumors studied herein.

Given the above, our objective is to examine the presence of intussusceptive phenomena, including those of intussusceptive angiogenesis and intussusceptive lymphangiogenesis, in several types of benign and malignant blood and lymphatic vessel tumors, as well as their participation in the growth and morphogenesis of these tumors. For this purpose, we assess (a) the types, structural components, connections, and arrangement of intussusceptive structures in benign vessel tumors, (b) the findings supporting the mechanisms in the formation of the intussusceptive structures, (c) intussusceptive structures in low-grade malignant vessel tumors, and (d) intussusceptive structures in malignant vessel tumors.

## 2. Material and Methods

### 2.1. Human Tissue Samples

Specimens (paraffin and epoxy resin blocks) of benign, low-grade malignant, and malignant vessel tumors were obtained from the archives of Histology and Anatomical Pathology of the Departments of Basic Medical Sciences of La Laguna University, University Hospital, and Eurofins^®^ Megalab–Hospiten Hospitals of the Canary Islands. The examples of benign cases included lobular capillary hemangioma (n: 12), intravascular papillary endothelial hyperplasia or Masson tumor (n: 10), sinusoidal hemangioma (n: 5), cavernous hemangioma (n: 8), glomeruloid hemangioma (n: 1), angiolipoma (n: 20), and lymphangiomas (n: 10). Angiolipoma was included in this group as an example of mature tissues with abnormal vessels. Examples of low-grade malignancy encompassed retiform hemangioendothelioma (n: 2) and papillary intralymphatic angioendothelioma or Dabska tumor (n: 1), while malignant tumors comprised angiosarcoma (n: 5) and Kaposi sarcoma (n: 5). From our previous studies in some vessel tumors, we randomly obtained cases of intravascular papillary endothelial hyperplasia [27], sinusoidal hemangioma [28], lymphatic malformations/lymphangiomas [29], and angiolipoma [32], selecting those of rapid evolution in lobular capillary hemangioma [30] and early stages in Kaposi sarcoma [33]. Ethical approval for this study was obtained from the Ethics Committee of La Laguna University [Comité de Ética de la Investigación y de Bienestar Animal, CEIBA 2023-3343], including the dissociation of the samples from any information that could identify the patient. The authors, therefore, had no access to identifiable patient information.

### 2.2. Light Microscopy

Samples were fixed in a 4% formaldehyde solution, embedded in paraffin, and serially sectioned at 3 µm. Sections were deparaffinized, hydrated, and alternatively stained with hematoxylin and eosin, and Masson trichrome (Roche, Basel, Switzerland. Ref. 6521908001).

In addition to the paraffin sections, a large number of semithin sections (1 µm) was obtained from samples previously fixed in a 2% glutaraldehyde solution with sodium cacodylate buffer (pH 7.4, for 6 h at 4 °C), post-fixed in 1% osmium tetroxide for two hours, and dehydrated and embedded in epoxy resin. Finally, the sections were stained with 1% Toluidine blue (Merck^®^, Darmstadt, Germany) and photographed using a Leica^®^ DM-750 light microscope (Wetzlar, Germany), with an integrated High-Definition Camera.

### 2.3. Immunohistochemistry and Immunofluorescence

For immunohistochemistry, two types of procedures were performed (automated and manual) as previously described [28,33]. The following primary antibodies were used in the automated procedure: anti-CD34 (Bond™ PA0212; Leica Biosystems, Newcastle, UK); anti-ERG (EPR3864, rabbit monoclonal, code No. 790-4576, Roche Diagnostics, S.L., Barcelona, Spain); anti-αSMA (Bond™ PA0943; Leica Biosystems, Newcastle, UK); and D2-40 monoclonal mouse anti-human and clone D2-40 (dilution 1:100) (Dako, Glostrup, Denmark), catalog No. M3619. For the manual procedure, mouse monoclonal anti-αSMA (1/100 dilution, code No. ABK1-A8914, Abyntek Biopharma, Zamudio, Spain) and rabbit polyclonal anti-CD34 (1/100 dilution, code No. A13929, ABclonal, Woburn, MA, USA) were used. A double immunofluorescence method was carried out using a combination of the following rabbit polyclonal antibodies: anti-CD34 (1/100 dilution, code No. A13929, ABclonal, Woburn, MA, USA) and anti-collagen type I (1/100 dilution, code No. AB749P, Millipore, Burlington, MA, USA). Anti-CD34 (mouse monoclonal antibodies anti-CD34, ready to use, class II, clone QBEnd 10, code No. IR632, Dako, Carpinteria, CA, USA) and anti-αSMA (1/100 dilution, code No. ABK1-A8914, Abyntek Biopharma, Zamudio, Spain). The following secondary biotinylated antibodies were then incubated for one hour at room temperature in the dark: goat anti-mouse IgG (1:300, Calbiochem, cat. No. 401213, Calbiochem, San Diego, CA, USA); Alexa Fluor 488 goat antirabbit IgG (H + L) (1:300, cat. No. A11001, Invitrogen, San Diego, CA, USA); biotinylated goat antirabbit IgG (H + L) (1:500, Code: 65-6140, Invitrogen); and Alexa Fluor 488 goat antimouse IgG (H + L) antibody (1:500, Code: A28175, Invitrogen). This was followed by incubation with streptavidin Cy3 conjugate (1:500, SA1010, Invitrogen) for one hour in the dark. Nuclei were stained with DAPI (Invitrogen, D1306, 1:10,000) for a five-minute incubation. Sections were coverslipped with DABCO (1%) and glycerol-PBS (1:1). Negative controls were performed in the absence of primary antibodies. Fluorescence immunosignals were obtained using a Fluoview 1000 laser.

### 2.4. Semiquantitative Analysis

A semiquantitative analysis of the frequency, characteristics of the cover and core, arrangement, and mechanisms of formation of pillars and meshes was performed of two of the authors. The parameters of frequency and characteristics of endothelial cells (endothelial cell morphology -flat/plump-, stratification, mitosis, atypia) were estimated as follows: 0 (absent, only for endothelial cells characteristics), 1 (<33%), 2 (33–66%), 3 (>66–<100%), and 4 (100%, referring to the tumor type with greater intensity of each estimated fact). The predominant core characteristics were as follows: 1: connective tissue; 2: fibrin; 3: complex structures; and 4: inflammatory component. The predominant arrangement of pillars and meshes were as follows: 1: densely grouped; 2: loosely grouped: 3: complex; 4: sinusoidal/linear; and 5: glomeruloid. The predominant mechanisms of formation were as follows: 1: vessel loops; 2: endothelialized thrombus; 3 intraluminal endothelial bridges; and 4: splitting and fusion.

## 3. Results

### 3.1. Intussusceptive Structures in Benign Vessel Tumors: Types, Structural Components, Connections, and Arrangement

#### 3.1.1. Types of Intussusceptive Structures in Benign Vessel Tumors

Intravascular/intraluminal pillars (less than or equal to 4 μm in diameter) and meshes (greater than 4 μm in diameter) (Figure 1A–C) were observed in the types of blood and lymphatic benign vessel tumors studied in the present work, including lobular capillary hemangioma or pyogenic granuloma, intravascular papillary endothelial hyperplasia or Masson tumor, sinusoidal hemangioma, cavernous hemangioma, glomeruloid hemangioma, angiolipoma, and lymphangiomas. Pillars (hallmarks of intussusceptive angiogenesis), oriented longitudinally, appeared and disappeared, depending on the view obtained in sequential views using confocal microscopy (Figure 1D).

#### 3.1.2. Structural Components of Meshes and Pillars in Benign Vessel Tumors

In vessel tumors, meshes and pillars presented a cover and a core. The cover was formed of endothelial cells, which were CD34 positive in blood vessel tumors (Figure 2A) and D2-40 positive in lymphatic vessel tumors (Figure 2B). The endothelial cells, arranged in a simple layer, were flattened or plump and without atypia or mitosis. The plump endothelial cells, either isolated or in small groups, were interspersed among the flattened cells. Packed collagen fibers and CD34+ stromal cells/telocytes were observed in the meshes and in the well-developed cores of the pillars in the blood and lymphatic vessel tumors (Figure 2A,B). Vascular smooth muscle cells or pericytes were also seen in the cores (immediately underlying the endothelial cover) of these intussusceptive structures in blood vessel tumors (Figure 2A). Depending on vessel tumor location, the core of wide meshes (tissue islands) could contain complex tissue structures, such as glands (Figure 3A), vessels (Figure 3B), nerves, smooth and striated muscles (Figure 3C,D, respectively), cutaneous annexes (Figure 3E), and bile ducts (Figure 3F).

#### 3.1.3. Connections Established by Intravascular Meshes and Pillars in Benign Vessel Tumors

Some areas of the cores of meshes and pillars (predominantly the ends) were observed in continuity with the media layer of the vessel wall and/or the perivascular tissue (Figure 4A). In these areas of the interstitial tissue structure connection, the endothelium of the vessel appeared curved, continuing with that which lines the folds and pillars (Figure 1B and Figure 4A). Continuities were also seen between the cores of different meshes or between those of meshes and pillars (Figure 4B).

Loose endothelial cover contacts, devoid of specific junctions, were frequently observed between meshes (Figure 4C) and between pillars, or between meshes and pillars. The endothelial cells in these contacting areas were frequently prominent.

#### 3.1.4. Arrangement of Meshes and Pillars in Benign Vessel Tumors

In general, meshes, with a round, elongated, or irregular morphology, were isolated and/or grouped (Figure 5A). The aggregation was linear (Figure 5B) or formed complex structures, some of which resembled segmented cactus with rounded cladodes (Figure 5C). Some meshes, aggregated linearly, were long and thinner and incompletely delimited vascular spaces together with the associated pillars, acquiring a sinusoidal appearance whose maximum expression occurred in sinusoidal hemangioma (Figure 5B). Grouped meshes and pillars in dilated vessels were observed, delimiting curved anastomosing channels, and resembled immature renal glomeruli (Figure 5D), which were the main structural component in the glomeruloid hemangioma (Figure 6A,B). When the meshes and pillars were grouped more densely, the luminal spaces between them were reduced (Figure 6C), so that in some areas they acquired a solid appearance, with narrow or virtual luminal spaces (Figure 6D).

### 3.2. Findings Supporting Different Mechanisms in the Formation of Meshes and Pillars in Vessel Tumors

The findings that contribute to the formation of meshes and pillars were mainly investigated in benign vessel tumors and pseudotumors. The main findings supporting different mechanisms were (a) thrombosis with endothelialization and cell invasion of the thrombotic material, (b) interendothelial bridges through the vessel lumen and subsequent incorporation of core components, (c) vessel loops surrounding interstitial tissue structures transported toward the vessel lumen, (d) splitting or fusion of meshes and meshes and pillars, pillars, and (e) combinations of the different findings mentioned above and/or with sprouting angiogenesis.

#### 3.2.1. Thrombosis with Endothelialization and Cell Invasion of the Thrombotic Material—The Thrombus as a Support of Vessel Intussusception

In some types of tumors, such as intravascular papillary endothelial hyperplasia and angiolipoma, thrombi or their fragments were observed with relative frequency in the vessel lumen (Figure 7A). Thrombotic components, including fibrin (Figure 7B), formed the initial core of meshes and pillars when they were covered by endothelial cells (Figure 7B). Thus, the fibrinous thrombus adhered to the vessel wall was endothelialized from endothelial cells in the intima of the vessel (primitive fold cover). In the border of the adherent region, the endothelial cells of the vessel wall appeared reoriented (curved), leaving the thrombus in continuity with the perivascular tissue (Figure 7C,D). In this area of continuity, stromal cells, pericytes, or vascular smooth muscle cells and inflammatory cells could be observed penetrating the thrombus (Figure 7E,F). All, or some, of these components, together with the extracellular matrix, including collagen, acquired the mature fold core characteristics. Meshes associated with pillars were observed dividing the vessel lumen (Figure 7G). Thrombi fragments were also seen adhering to meshes, and new meshes and pillars were formed from them by a similar procedure.

#### 3.2.2. Interendothelial Bridges through the Vessel Lumen and Subsequent Incorporation of Core Components 

Interendothelial bridges (nascent pillars) and mature pillars were observed between opposite vessel walls, between meshes and the vessel wall, and between meshes (Figure 8A–C). In the sections, the nascent pillars showed endothelial cell bilayers with virtual cores. Thin extensions of pericytes or their processes and/or stromal cells, as well as collagen fibers, were also seen starting to penetrate the endothelial cell bilayer in other evolving nascent pillars (Figure 8A,B). Finally, well-formed pillars presented their characteristic cover and well-developed core (Figure 8C).

#### 3.2.3. Formation of Vessel Loops and Transport of the Surrounded Interstitial Tissue Structures toward the Vessel Lumen—Dissecting Phenomenon—Piecemeal Form of Intussusceptive Angiogenesis

Vessel loops formed from preexisting vessels or from other loops (secondary loops), surrounding interstitial tissue structures, were frequently observed in vessel tumors, being very evident in lymphangiomas and intravascular papillary endothelial hyperplasia. When the loop lumen was open, the endothelial cell layer of the vessel loop that surrounds the interstitial tissue structure (forming the mesh cover) and the interstitial tissue structure (forming the mesh core) appeared to be projecting into (transported to) the lumen of the preexisting vessel fused with that of the loop (Figure 8D). Some of the meshes formed by this procedure were exceptionally large (giant meshes or tissue islands) and most frequently showed vessels, nerves, muscle, glands, and/or skin annexes in their cores (Figure 8D). Vessels and inflammatory infiltrate were observed to be partially transported to the vessel lumen in Kaposi sarcoma, giving rise to the promontory sign (Figure 8E).

#### 3.2.4. Splitting and Association of Meshes

Folded endothelial cells splitting meshes (Figure 1A) and fusion or association (Figure 2A and Figure 4C) of meshes were observed in the vessel tumors studied.

#### 3.2.5. Combinations of the Different Findings Mentioned above and/or with Sprouting Angiogenesis

Combinations of the different findings leading to formation of meshes and pillars could be seen depending on the tumor region examined and even in the same region. Findings of sprouting angiogenesis associated with those of intussusceptive angiogenesis were also observed and were highly expressive in the lobular capillary hemangioma (pyogenic granuloma). In this association, endothelial cells sprouting from preexisting vessels or from other loops were seen forming loops which surrounded interstitial tissue structures. In this process, the loop lumens frequently remained very narrow, and intravascular meshes and pillars were more difficult to identify, only being evident when the loop lumens were sufficiently open (Figure 9).

### 3.3. Intussusceptive Structures (Meshes and Pillars) in Low-Grade Malignant Vessel Tumors (Locally Aggressive/Rarely Metastasizing)

Intraluminal meshes and pillars were observed in some ectatic vessels of the arborizing vascular channels of the retiform hemangioendothelioma. The endothelial cells that line the vessels, meshes, and pillars showed a hobnail or matchstick morphology (nucleus located apically forming a surface bulge) (Figure 10A,B). In the Dabska tumor (papillary intralymphatic angioendothelioma), meshes, covered by aggregates of voluminous CD34 and D2-40 positive endothelial cells, with high nuclear cytoplasmic ratio and moderate mitotic activity, appeared as tufts (Figure 10C–G) and could adopt a glomeruloid aspect.

### 3.4. Meshes and Pillars in Malignant Vessel Tumors

In the vasoformative pattern of angiosarcoma, the endothelial cells of meshes and pillars ranged from well-differentiated to atypical. In general, the endothelial cells increased in number and became prominent, and the hyperchromatic nuclei, ERG protein positive, became enlarged (Figure 11A,B). Intravascular meshes and pillars delimited spaces, which appeared as vessel-like structures (Figure 11C). These spaces were more densely arranged between them in the diagnostic areas of the lesion, acquiring the aspect of anastomosing vascular channels (Figure 11D). The dissecting phenomenon and the cores of meshes presenting compact collagen were evident. Complex structures could also be seen in the cores. Meshes and pillars were difficult to detect in the solid pattern of angiosarcoma formed of masses of endothelial cells with an epithelioid or fusiform appearance. However, in the mixed pattern, the numbers of layers of neoplastic endothelial cells covering the mesh and pillar cores progressively increased in the transitional areas (Figure 12A,B), acquiring a solid pattern, while the fold and pillar cores gradually became less evident (Figure 12C,D).

In the patch and plaque stages of Kaposi sarcoma, meshes and pillars were evident in the lumen of neovessels. Only in plaque stages, the endothelial cells in these intussusceptive structures showed sparse mitotic figures, without nuclear pleomorphism. Pre-existing blood vessels, with or without perivascular inflammatory infiltrate, were present in many cores of the meshes, giving rise to the promontory sign (vessels within vessels) mentioned above (Figure 8E). In the nodular stage of Kaposi sarcoma, meshes and pillars were not evident.

### 3.5. Common and Changing Events in Pillars and Cores in Benign and Malignant Blood and Lymphatic Vessel Tumors

Common and changing events in pillars and meshes were observed in the benign and malignant blood and lymphatic vessel tumor samples included in the present study, although the changing findings could be combined in the same type of tumor. The common findings, especially the presence of these intussusceptive structures in all vessel tumor types, were very important in supporting the involvement of intussusception in vessel tumors. The changing events (frequency, cover and core characteristics, arrangement, and predominant formation mechanisms of intussusceptive structures) varied according to tumor type and were frequently intermingled in the same tumor. Intravascular papillary endothelial hyperplasia was the vessel tumor (pseudotumor, reactive lesion) with the highest number of pillars and meshes (myriad) and variations in their mechanisms of formation, while transitional areas between vasoformative and solid patterns of angiosarcoma were those with a higher number of mitosis and atypia in the endothelial cells that formed the covers of pillars and meshes. In Table 1, we summarize these findings.

## 4. Discussion

In several types of blood and lymphatic vessel tumors, including benign and malignant examples, we report the presence of intussusceptive angiogenic and lymphangiogenic morphologic structures, comprising pillars and meshes. In these intussusceptive structures in vessel tumors, we will consider (a) the distinction, characteristics, contacts, arrangement, morphogenic role, and possible formation mechanisms, (b) the differences depending on tumor benignity or malignance, and (c) the coincidences and relationship with other normal intravascular structures.

In our observations, intussusceptive structures are present in all the vascular tumors studied, and we highlight the few contributions in this regard. Pillars are difficult to identify with conventional optical microscopy, but meshes are not. Pathologists refer to these identifiable, thicker intussusceptive structures in vessel tumors as papillae, giving them a classificatory meaning in diagnostic practice. However, their intussusceptive relationship has rarely been considered. 

For the distinction between pillars and meshes, their diameter has generally been considered thus: pillars ≤2.5 μm and meshes ≥2.5 μm [5]. In our study, we have followed authors who consider 4μm as the limit between them [19,20], although others accept up to 5 μm [24]. Since pillars are the hallmark of intussusceptive angiogenesis, and their precise identification requires 3D demonstration, we used confocal microscopy to obtain sequential views comprising the thickness of longitudinally oriented pillars, which allowed us to verify that they met the characteristics to be considered as such. In addition, the structural characteristic of pillars, meshes, and folds in vessel tumors, presenting a cover formed of endothelial cells and a core formed of connective tissue components (αSMA+ pericytes/vascular smooth muscle cells, CD34+ stromal cells/telocytes, myofibroblasts, and collagen I) are similar to those described by different authors in several pathophysiological conditions [4,5,10,15]. 

The presence of loose contacts of the endothelial cover between meshes, pillars, and folds is suggestive of unstable junctions. These loose contacts may facilitate sudden changes, including quick division and adaptation of vessels, which concurs with the observations in other studies, mainly in a recent one in which in vivo in 3D with high spatio-temporal resolution was used [19]. To the contrary, the existence of continuities between the cores of meshes, and meshes and pillars, may entail greater stability, with more permanent arrangements of these structures, whose different appearance (e.g., papillary, sinusoidal, glomeruloid) conditions the morphologic expression of the lesions and subsequent classification of many of the vessel tumor types. Therefore, intussusception has an important role in vessel tumor morphogenesis.

In some benign and malignant vessel tumors, meshes, pillars, and folds can go unnoticed. This is mainly due to their dense grouping and the decreasing size of luminal spaces between them, which occurs in lobular capillary hemangioma, in which, as its name indicates, they acquire a lobular appearance, and in angiosarcoma, in which the interposed spaces adopt an aspect of anastomosing vascular channels. 

A major limitation to following the formation of intussusceptive structures was that observations were made under microscopic static conditions. To overcome (minimize) these difficulties, we consider meshes, pillars, folds as having characteristics suggestive of being in a different evolutionary stage and corresponding to different mechanisms of formation. In addition, to better follow the possible sequence, we selected cases with rapid development and spontaneous regression phenomena, mainly in lobular capillary hemangioma. Next, we summarize these possible mechanisms in vessel tumors, pinpointing those in which the principal findings are vessel loop formation and transport of the surrounded interstitial tissue structures toward the original lumen of the parent vessel.

(1)The mechanism of vessel loop formation and transport of the surrounded interstitial tissue structures toward the original lumen of the parent vessel can be considered a dissecting phenomenon or a piecemeal form of intussusceptive angiogenesis. The following sequence is possible: (a) endothelial cells (future cover of intussusceptive structures) sprouting from preexisting/parent vessels form vessel loops, with virtual lumen, surrounding interstitial tissue structures (future cores of intussusceptive structures), (b) when the vessel loop lumen—connected at both ends with the parent vessel—becomes patent, the interstitial tissue structure and surrounding endothelial cells are pushed toward the parent vessel lumen, and (c) as the lumen of loops increases and merges with that of the parent vessel, the intussusceptive structures appear to be transported to the common lumen. The most evolved findings of this process were identifiable in all vessel tumors, including Kaposi sarcoma (giving rise to the promontory sign) [33]. The earlier events were best detected in lobular capillary hemangioma, a lesion of rapid development in some cases. Taken together, the observations show that sprouting angiogenesis and intussusceptive angiogenesis associate and behave as complementary mechanisms with synergistic interaction [7,17,18,19,20,30,34,35,36] and that sprouting events precede the intussusceptive. This also coincides with experimental observations in neo-vascularized femoral vein walls [17] and with the most recent contributions on the remodeling of the zebrafish caudal vein plexus [19].(2)In the mechanism of thrombosis with endothelialization and cell invasion of the thrombotic material, thrombus components form a provisional core of intussusceptive structures. The following sequence is possible: (a) thrombus fragments adhere to the vessel wall or to previous meshes, (b) endothelial cells reorient their trajectory toward the surface of the thrombotic material from the ends of the adherent region, which becomes deendothelialized, (c) endothelialization of the surface of the thrombus fragments occurs from the reoriented endothelial cells (cover formation), and (d) penetration through the deendothelialized adherent region of connective cells and formation of extracellular matrix (mature core formation). Thrombosis has been related with intussusceptive angiogenesis in COVID-19 disease [37]. The mechanism posited here coincides with one described in experimental studies in blood vessels of the ovarian pedicle [4,5] and in hemorrhoidal disease [14]. In our observations, it was plainly evident in intravascular papillary endothelial hyperplasia and angiolipoma.(3)The mechanism of intraluminal endothelial bridge formation is similar to that traditionally outlined in initial contributions on intussusceptive angiogenesis, currently known as fusion of the intraluminal capillary wall [19], and to that described as inverse sprouting [13], although in vessel tumors, pillars mainly extend between meshes, and between meshes and the vessel wall. The following sequence is possible: (a) the endothelial bridges, forming nascent pillars (initial pillar cover), extend between the opposite vessel wall, between meshes, and between meshes and the vessel wall, and (b) pericytes and stromal cell extensions penetrate endothelial bridges and collagen is formed or transported, originating from the pillar core. Although the findings that support this mechanism for intussusceptive formation coincide with those described in other physio-pathological and experimental conditions [1,2,3,4,5,6,7,8,9,10,11,12,13,14,38], our observations suggest that, in vessel tumors, it only occurs in thin pillars.(4)In the splitting and fusion of intussusceptive structures in vessel tumors, the formation of new intussusceptive structures through splitting occurs in meshes through the folding of their endothelial cells, a process also described in several conditions [15]. Merged or simply associated structures give rise to the more complex formations mentioned above, depending on their arrangement.

Interestingly, the morphology, arrangement, proliferative activity, and typical or atypical characteristics of endothelial cells that form the cover of meshes, pillars, and folds are different in vessel tumors depending on their benignity (flattened, with some plump intercalated endothelial cells, arranged in a single layer, without mitosis or atypia), low-grade malignancy (with a hobnail/matchstick or plump morphology, high nuclear to cytoplasmic ratio, arrangement in one or several layers, or forming aggregates, and moderate mitotic activity), or malignancy (plump, atypical, multilayered, or forming large aggregates and with atypical mitoses). These differences arise because the cover of intussusceptive structures originates from malignant endothelial cells in cancerous vessel tumors, unlike in benign vessel tumors or in vessels of nonvascular tumors. Conversely, changes in the cores of pillars, meshes, and folds, depending on benignity or malignancy, were less observed, except in solid malignant vessel tumors, in which they become inconspicuous.

Many vessel tumors/pseudo-tumors develop from veins and lymphatics in which valves form. We have observed that morphological events during valve development are similar to those of intussusceptive structures, including the formation of their principal components: endothelial cover and connective core, which points to intussusceptive mechanisms participating in valve development (unpublished observations). Likewise, fluid flow-associated forces have been shown to participate in regulating molecules involved in valve development and position, particularly in vessel branch points [39,40,41]. A similar influence on intussusceptive phenomena of the hemodynamic conditions, including shear stress and changes in blood flow intensity, velocity, and pressure [3,7,12,18,42,43,44,45,46,47,48,49,50,51], as well as hypoxia [52,53], have been shown to control vessel intussusceptive events through molecular signals, including VEGF [54,55], nitric oxide [56], Ang-2, endoglin [57], ephrinB2/EphB4 signaling [58], and endothelial cell MT1-MMP and Notch signaling [59,60]. In recent observations, although the mechanisms of transluminal pillar formation were not directly associated with blood flow in the zebrafish caudal vein plexus, impairment of mesh formation in the absence of blood flow was observed [19]. The induced position of valves in vessel branch points also coincides with that described for intussusception [45]. Similar intussusceptive phenomena have also been demonstrated in the meshwork formation of intraluminal processes in the sinuses of developing lymph nodes [61], in which the influence of fluid flow and molecular control also participate [61,62,63,64]. Consequently, intussusceptive phenomena in vessel tumors could be the pathological counterpart of intravascular development of normal intussusceptive structures and their regulatory systems in blood and lymphatic vessels.

Our contributions in vessel tumors, highlighting the participation of intussusceptive angiogenesis in the morphogenesis and in vascular network growth, can provide the groundwork for future studies on regulatory molecular mechanisms of intussusception in these diseases and their possible control.

## 5. Conclusions

In several types of benign and malignant blood and lymphatic vessel tumors, we observe the presence of pillars (<4 μm), meshes (≥4 μm), and folds, which are the morphological expression of intussusceptive angiogenesis that participate, together with sprouting angiogenesis, in vessel network growth and morphogenesis. In vessel tumors, the intussusceptive structures showed (1) an endothelial cover and a connective core; (2) loose junctions and different arrangements, which conditioned the vessel tumor morphology (substrate of vessel tumor type), sometimes with dense grouping that leads to their being overlooked; (3) findings that suggested several and frequently associated mechanisms in their development, including (a) formation of vessel loops and transport of the surrounded interstitial tissue structures toward the original lumen of the parent vessel, (b) thrombus components as a provisional core, (c) formation of intraluminal endothelial bridges, and (d) splitting and fusion of previous intussusceptive structures; and (4) different morphology, arrangement, proliferative activity, and typical or atypical characteristics of endothelial cells in their covers depending on the vessel tumor benignity, low-grade malignancy, or malignancy. Intussusceptive structures in vessel tumors could be the pathological counterpart of the normal intraluminal structures (e.g., valves, intraluminal processes in the sinuses of developing lymph nodes). Our contributions lay the foundations for more in-depth molecular studies based on intussusception in vessel tumors, which entail therapeutic strategies, and are therefore of morphogenic, clinical, and therapeutic interest. 

## Figures and Tables

**Figure 1 biomedicines-12-00258-f001:**
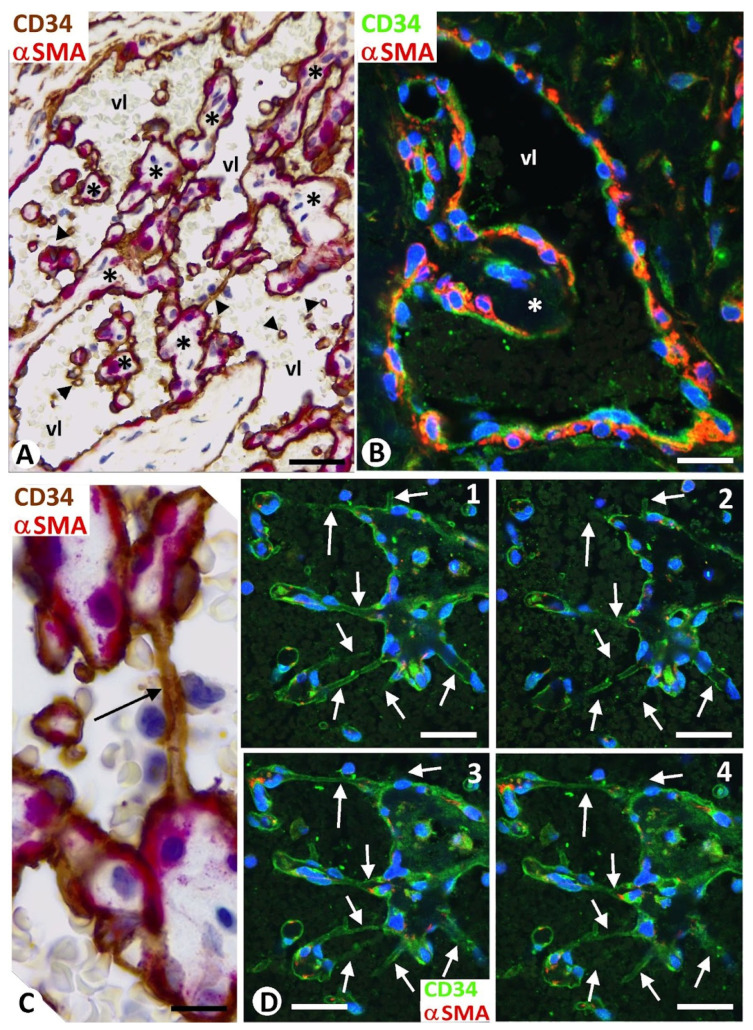
(**A**): Intravascular meshes (asterisks, ≥4 μm diameter), and pillars (arrowheads, ≤4 μm diameter), transversally and obliquely sectioned. (**B**): At higher magnification, an initial mesh (asterisk). (**C**): A longitudinally sectioned pillar (arrow) between two meshes. Note that the meshes and pillars are covered by endothelial cells (brown in (**A**,**C**) and green in (**B**)) and that αSMA-positive mural cells (red) are arranged underlying the endothelial cells. (**D**): Sequential views in confocal microscopy showing the appearance and disappearance of several longitudinally sectioned pillars (arrows) in a 4 μm tissue section. Images from intravascular papillary endothelial hyperplasia. Vessel lumen: vl. (**A**,**C**): Double immunochemistry for CD34 (brown) and αSMA (red). (**B**,**D**): Double immunofluorescence for CD34 (green) and αSMA (red). Bar: (**A**): 50 μm; (**B**–**D**): 10 μm.

**Figure 2 biomedicines-12-00258-f002:**
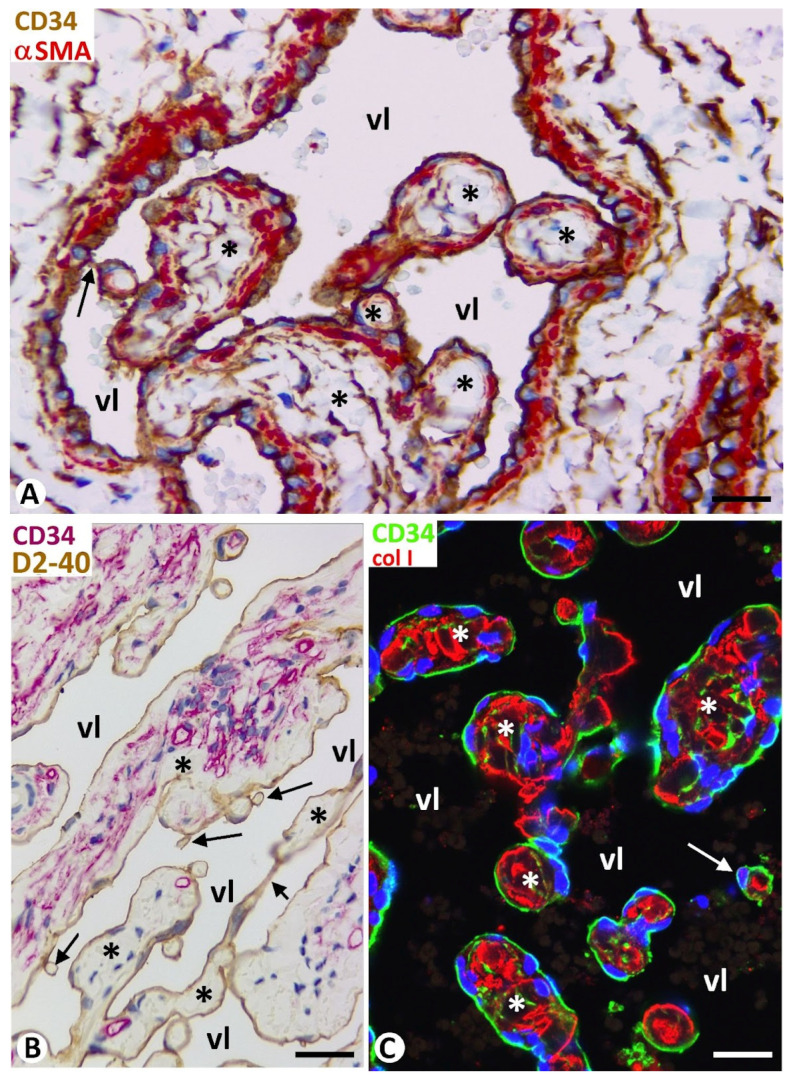
(**A**–**C**): Structural components of meshes (asterisks), and pillars (arrows), each showing a cover and a core. Note that the cover is formed of endothelial cells CD34+ (brown in (**A**) and green in (**C**)) in cavernous hemangiomas and D2-40+ (brown in (**B**)) in a cutaneous lymphangioma. αSMA+ pericytes (red in (**A**)), CD34+ stromal cells (brown in (**A**) and red in (**B**)), and collagen I (red in (**C**)) are shown in the cores of meshes and pillars. (**A**): Double immunochemistry for CD34 (brown) and αSMA (red). (**B**): Double immunochemistry for D2-40 (brown) and CD34 (red). (**C**): Double immunofluorescence for CD34 (green) and collagen I (red). Bar: (**A**,**C**): 10 μm; (**B**): 20 μm.

**Figure 3 biomedicines-12-00258-f003:**
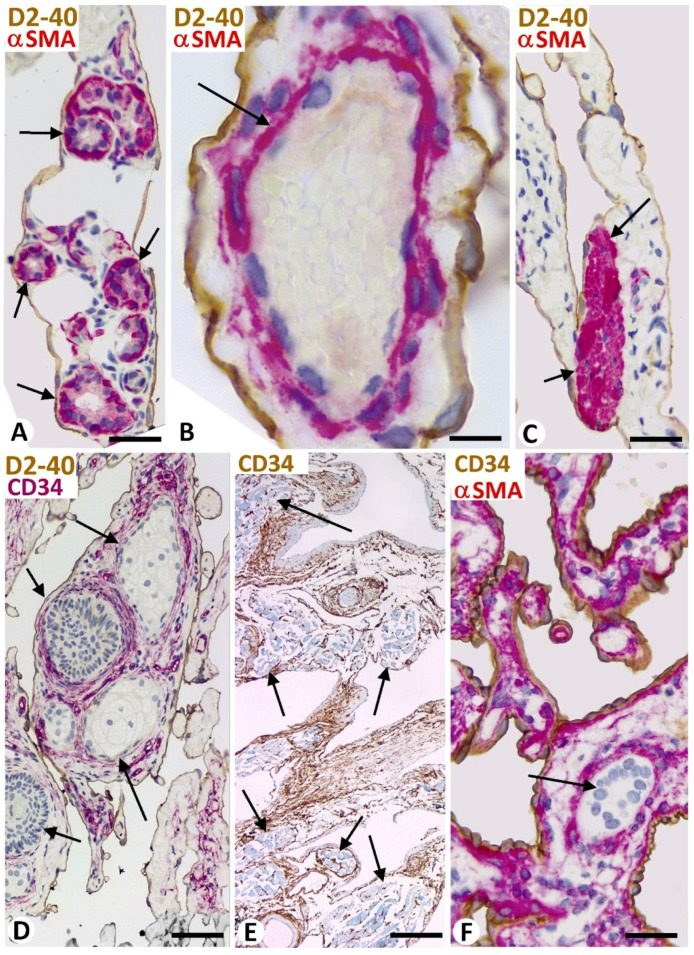
(**A**–**D**): Complex tissue structures in the core of large (giant) meshes (tissue islands) in cutaneous lymphangiomas (**A**–**D**), including sweat glands (arrows in (**A**)), a vessel (arrow in (**B**)), arrector pili smooth muscle (arrows in (**C**)), and hair and sebaceous glands (arrows in (**D**)). (**E**): Striated muscle (arrows) in the core of large meshes and folds in a lingual lymphangioma. (**F**): A bile duct (arrow) in the core of a large mesh in a cavernous hepatic hemangioma. (**A**–**C**): Double immunochemistry for D2-40 (brown, showing endothelial cells) and αSMA (red, showing myoepithelial cells in (**A**), vascular mural cell in (**B**), and smooth muscle in (**C**)). (**D**): Double immunochemistry for D2-40 (brown, showing endothelial cells) and CD34 (red, showing stromal cells). (**E**): Immunochemistry for CD34 (brown, showing stromal cells). (**F**): Double immunochemistry for CD34 (brown, showing endothelial cells) and αSMA (red). Bar: (**A**,**C**,**D**): 40 μm; (**B**): 15 μm; (**E**): 100 μm; (**F**): 20 μm.

**Figure 4 biomedicines-12-00258-f004:**
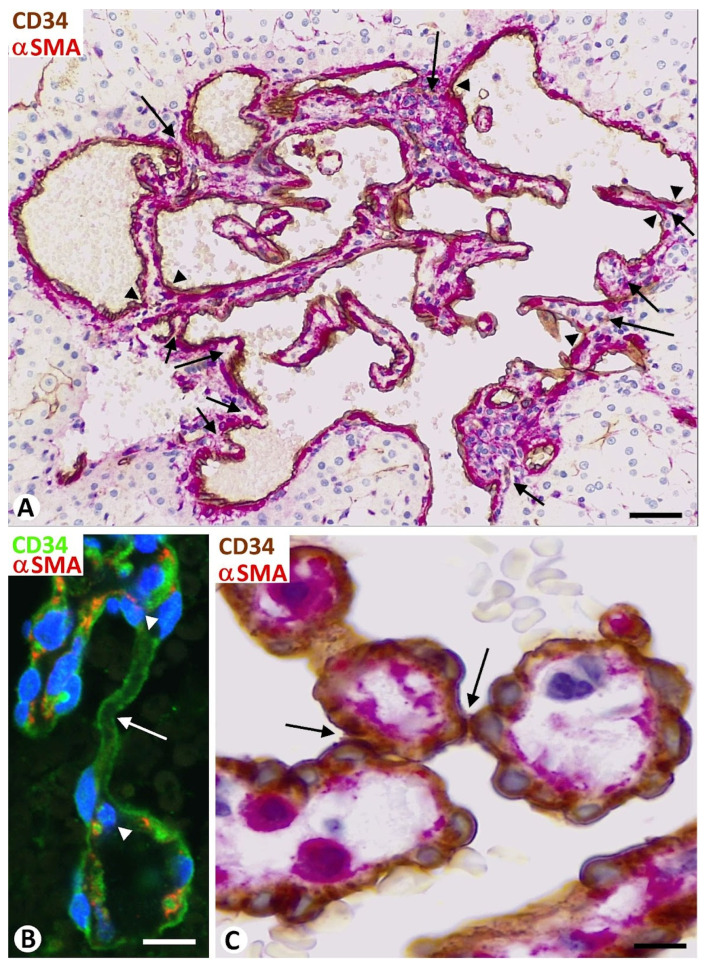
Connections established by intravascular meshes and pillars in benign vessel tumors. (**A**): Continuities of the cores of intravascular meshes with the perivascular tissue (arrows). Note that the vessel endothelium curves in the areas of interstitial tissue structure connection (arrowheads). (**B**): A pillar (arrow) between two meshes with continuity of the cores at pillar ends (arrowheads). (**C**) Endothelial cover contacts (arrows) are seen between several meshes. (**A**,**C**): Double immunochemistry for CD34 (brown) and αSMA (red). (**B**): Double immunofluorescence for CD34 (green) and αSMA (red). Bar: (**A**): 35 μm; (**B**,**C**): 15 μm.

**Figure 5 biomedicines-12-00258-f005:**
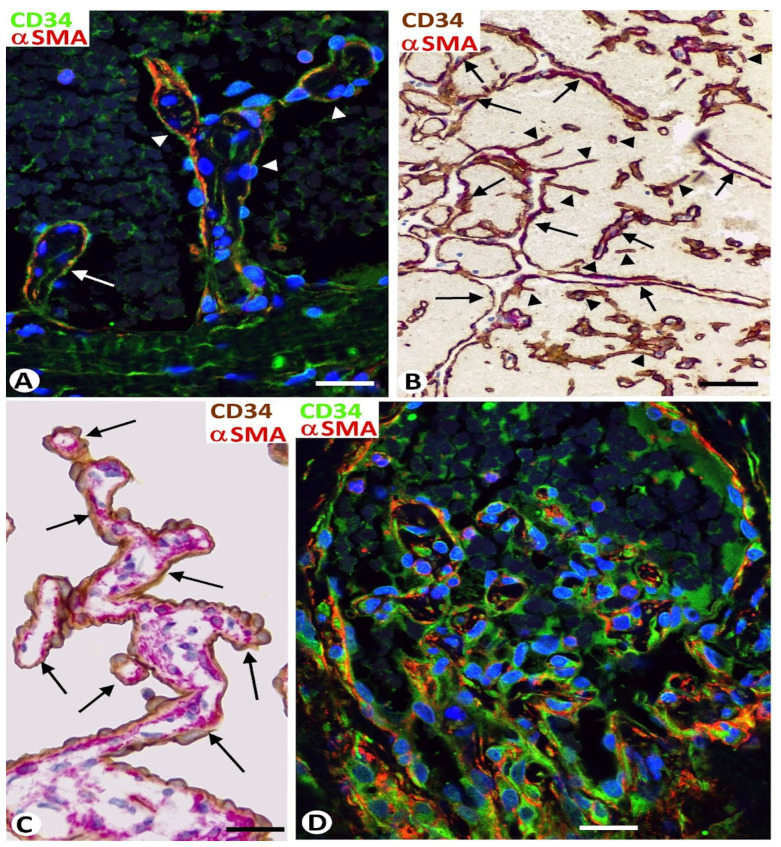
Arrangement of meshes and pillars in benign vessel tumors. (**A**): Isolated (arrow) and aggregated (arrowhead) meshes are observed in the lumen of a vessel. (**B**): Thin meshes (arrows) and associated pillars (arrowhead), with predominant linear arrangement, incompletely delimit vascular spaces (sinusoidal appearance). (**C**): Grouped meshes (arrows) resembling segmented cactus with cladodes. (**D**): Meshes and pillars in a dilated vessel resembling an immature renal glomerulus. Double immunofluorescence (**A**,**D**) and double immunochemistry (**B**,**C**) for CD34 (green in (**A**,**D**) and brown in (**B**,**C**) showing endothelial cells) and αSMA (red, showing pericytes). Bar: (**A**,**C**): 20 μm; (**B**): 50 μm, (**D**): 35 μm.

**Figure 6 biomedicines-12-00258-f006:**
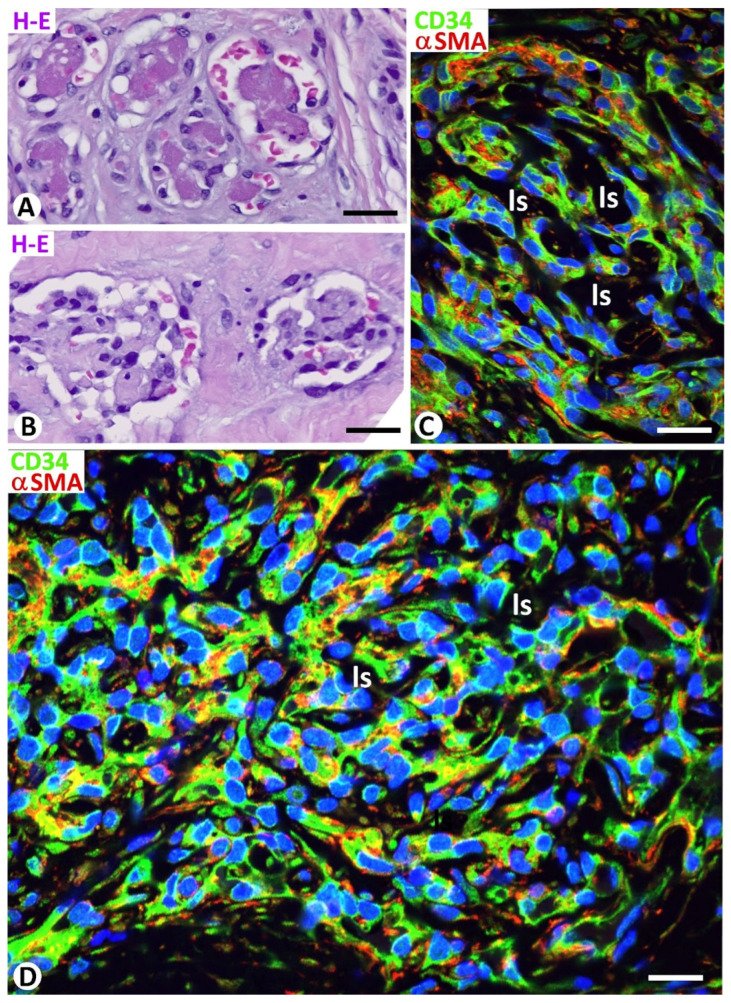
(**A**,**B**): Immature (**A**) and mature (**B**) glomeruloid structures in a glomeruloid hemangioma. (**C**,**D**): Meshes and pillars densely grouped, showing a cover formed of endothelial cells (green) and a core with pericytes (red). Note that the luminal spaces (ls) between the intussusceptive structures are reduced (**C**) and become very narrow or virtual (solid appearance) (**D**). (**A**,**B**): Hematoxylin and eosin staining. (**C**,**D**): Double immunofluorescence for CD34 (green) and αSMA (red). Bar: (**A**,**B**): 50 μm; (**C**): 30 μm; (**D**): 12 μm.

**Figure 7 biomedicines-12-00258-f007:**
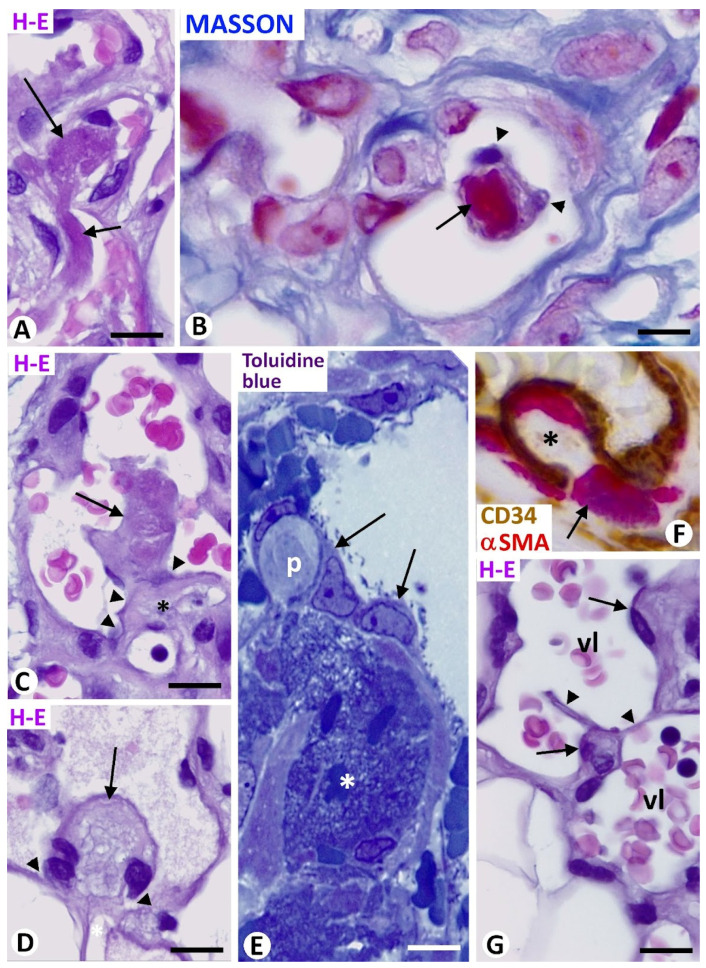
Thrombi as support for formation of meshes and pillars. (**A**,**B**): Thrombi in small vessels (arrows). Note, in (**B**), fibrinous material of the thrombus (initial core, red) covered by endothelial cells (arrowheads). (**C**,**D**): Evolutive thrombi (arrows) adhered to the vessel walls. Observe the reoriented (curved) endothelial cells in the border of the adherent region (arrowhead), leaving the thrombus in continuity with the perivascular tissue (asterisk). (**E**): A pillar (p) is seen originating from a fibrinous thrombus (asterisk) covered by endothelial cells (arrows). (**F**): A mesh (asterisk) in which a pericyte (red, arrow) is observed penetrating the core covered by endothelial cells (brown). (**G**): Meshes (arrows) and pillars (arrowheads) are seen dividing the vessel lumen. (**A**,**C**,**D**,**G**): Hematoxylin and eosin staining. (**B**): Masson trichrome staining. (**E**): Semithin section. Toluidine blue. (**F**): Double immunochemistry for CD34 (brown) and αSMA (red). Bar: (**B**,**E**): 10 μm; (**A**,**C**,**D**,**F**,**G**): 15 μm.

**Figure 8 biomedicines-12-00258-f008:**
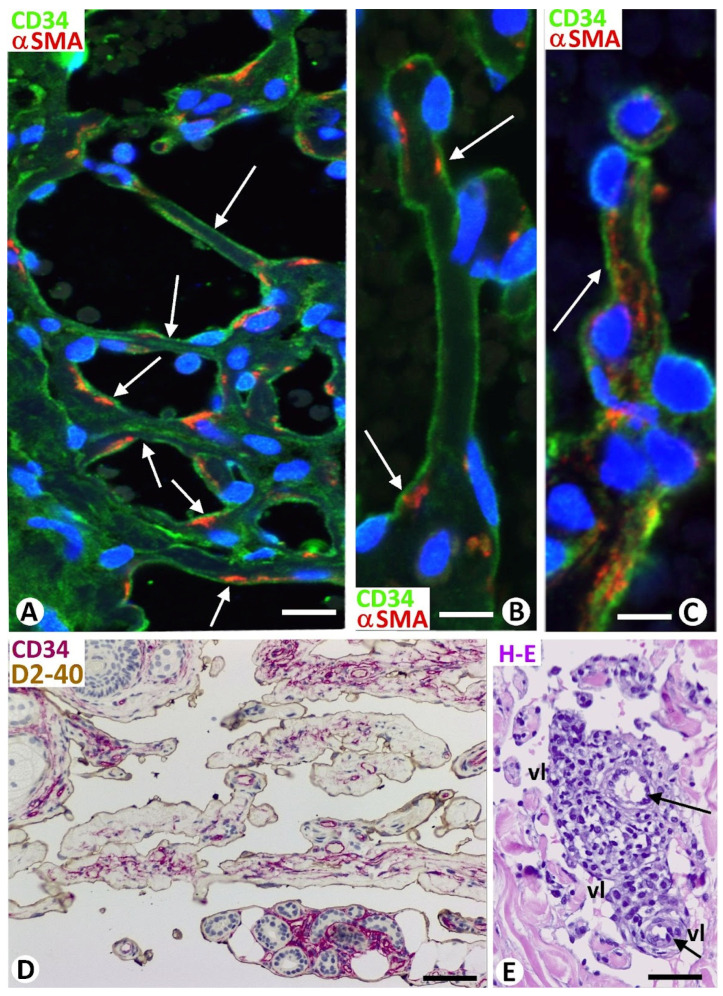
(**A**–**C**): Formation of pillars by interendothelial bridges through the vessel lumen and subsequent incorporation of core components. (**A**,**B**): Longitudinally sectioned nascent pillars with a double layer of endothelial cells (green) between which pericytes or their processes (red) begin to appear. (**C**): A mature pillar (arrow) with a cover formed of endothelial cells (green) and a core showing pericytes (red). (**D**,**E**): Formation of vessel loops and transport of the surrounded interstitial tissue structures toward the vessel lumen. (**D**): Intussusceptive structures, with interstitial tissue components forming their cores, in which stromal cells are observed (red), and endothelial cells forming their covers (brown), appear transported in the vessel lumen (dissecting phenomenon) in a cutaneous lymphangioma. Note the presence of skin annexes in some of these thick intussusceptive structures. (**E**): Blood vessels (arrows) surrounded by inflammatory infiltrate are partially incorporated into the vessel lumen (vl) of a Kaposi sarcoma, giving rise to the promontory sign. (**A**–**C**): Double immunofluorescence for CD34 (green) and αSMA (red). (**D**): Double immunochemistry for D2-40 (brown) and CD34 (red). (**E**): Hematoxylin and eosin staining. Bar: (**A**): 20 μm; (**B**,**C**): 10 μm; (**D**): 60 μm; (**E**): 50 μm.

**Figure 9 biomedicines-12-00258-f009:**
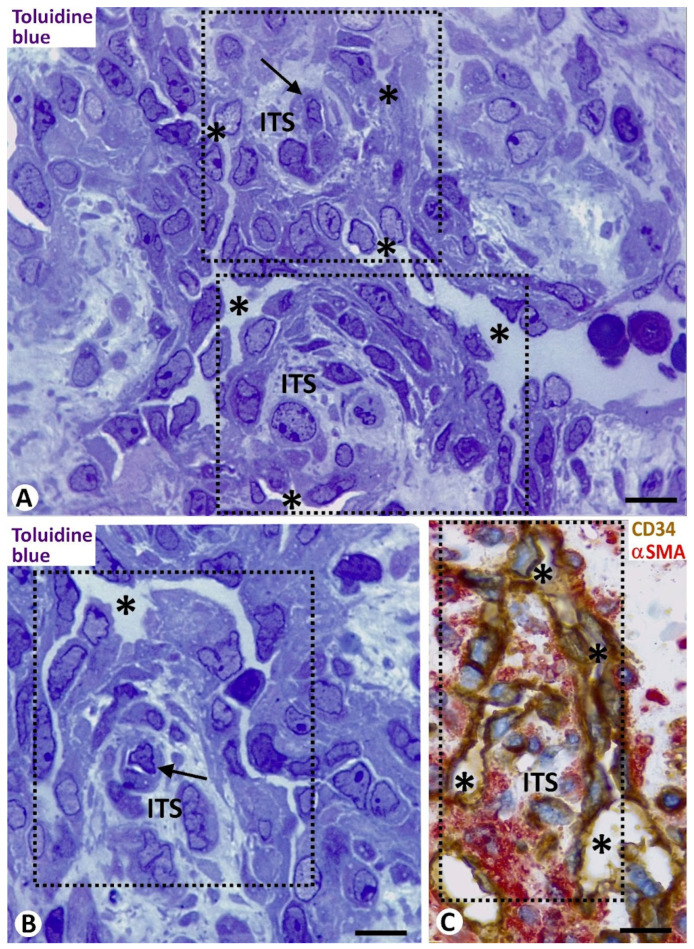
Findings of sprouting angiogenesis associated with those of intussusceptive angiogenesis in lobular capillary hemangioma (pyogenic granuloma). Interstitial tissue structures (ITS) surrounded by endothelial cell sprouts with thin lumens (asterisks), virtual on occasions. Note in boxes that the surrounding endothelial cells of the sprouts and the surrounded ITSs form meshes and that the ITSs can contain vessels (arrows). (**A**,**B**): Semithin sections. Toluidine blue staining. (**C**): Image of a mesh formed of an interstitial tissue structure (the core), showing αSMA+ pericytes (red), and covered by the CD34+ internal endothelial layer (brown) of a loop (double immunochemistry for CD34 and αSMA). Bar: (**A**,**B**): 10 μm; (**C**): 15 μm.

**Figure 10 biomedicines-12-00258-f010:**
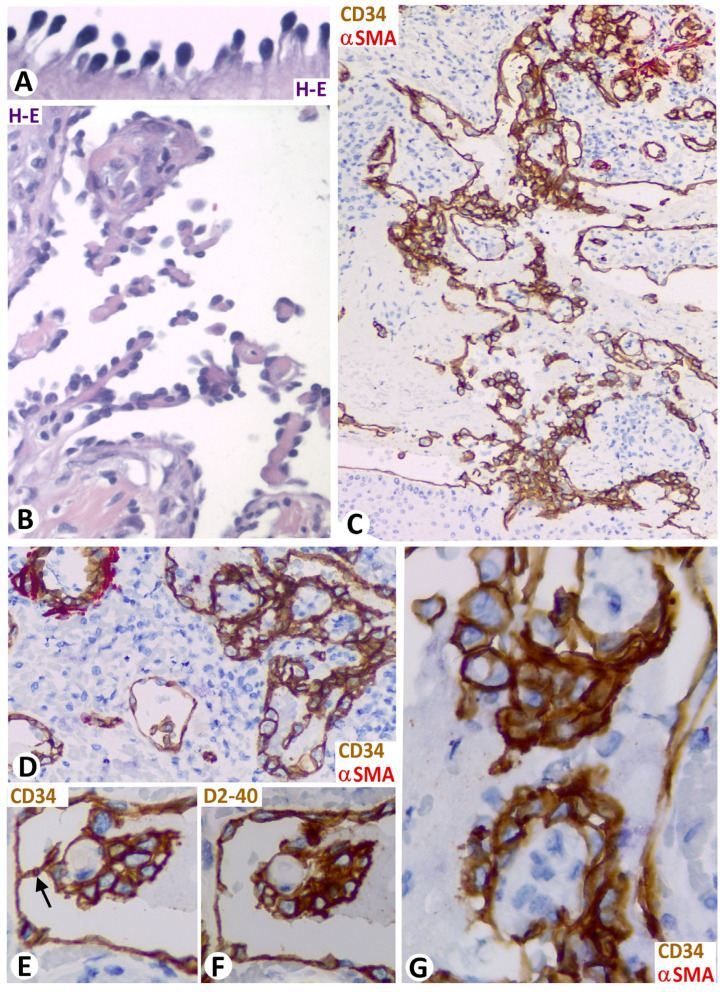
(**A**,**B**): Retiform hemangioendothelioma in which the endothelial cells that line a dilated vessel (**A**) and meshes and pillars (**B**) show nuclei in their superficial region of the cells, adopting a hobnail or matchstick appearance. (**C**–**G**): Dabska tumor in which meshes. (**E**,**F**) correspond to two contiguous sections of the same specimen, in which endothelial cells show expression for CD34 (**E**) and podoplanin (**F**), respectively. Note a pillar (arrow) in (**E**) that disappears in (**F**). (**A**,**B**): Hematoxylin and eosin staining. (**C**,**D**,**G**): Double immunochemistry for CD34 and αSMA. E: Immunochemistry for CD34. (**F**): Immunochemistry for D2-40. Bar: (**A**): 20 μm; (**B**,**C**): 35 μm; (**D**–**G**): 20 μm.

**Figure 11 biomedicines-12-00258-f011:**
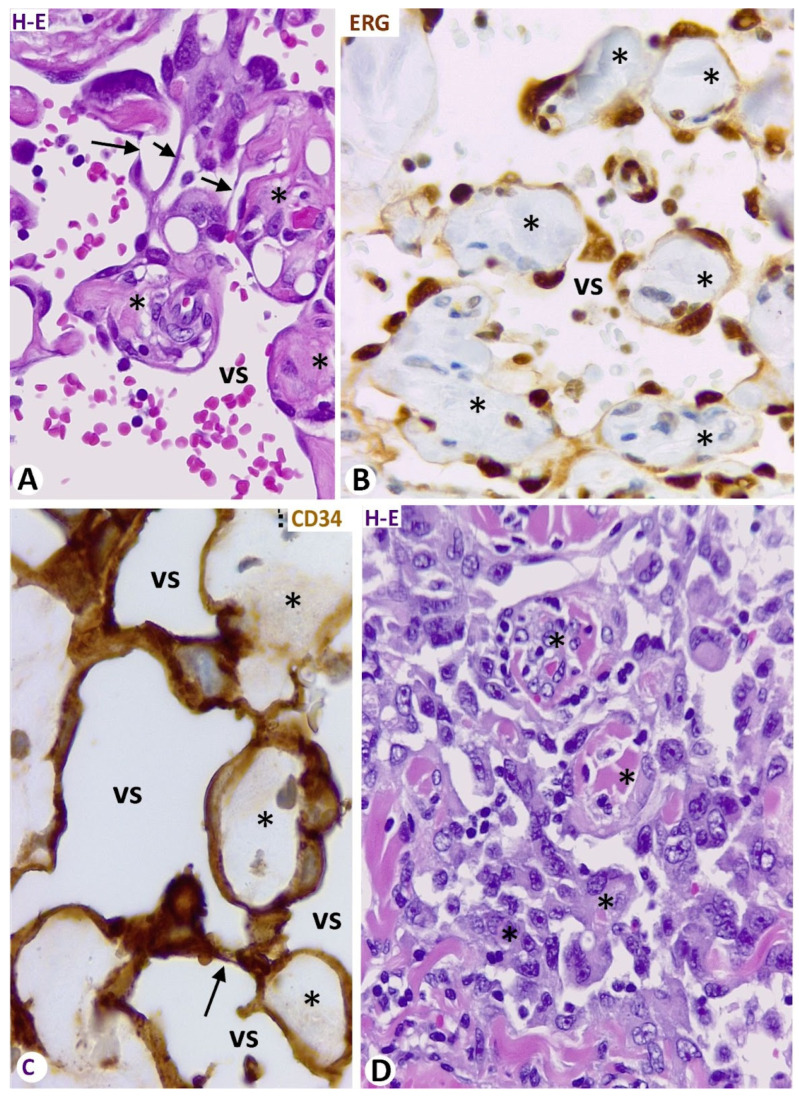
Meshes (asterisk), and pillars (arrows) in the vasoformative pattern of angiosarcoma. (**A**): Nuclear atypia is observed in the lining endothelial cells. (**B**): ERG protein expression in the atypical nuclei of the lining endothelial cells (brown). (**C**): Vessel-like spaces (vs) appear delimited by the lining endothelial cells (CD34+, brown) of connecting meshes and pillars. (**D**): A diagnostic area of an angiosarcoma in which the lining atypical endothelial cells of more grouped connecting meshes and pillars delimit spaces that acquire the aspect of anastomosing vascular channels. (**A**,**D**): Hematoxylin and eosin staining. (**B**): Immunochemistry for ERG protein (brown). (**C**): Immunochemistry for CD34 (brown). Bar: (**A**–**D**): 10 μm.

**Figure 12 biomedicines-12-00258-f012:**
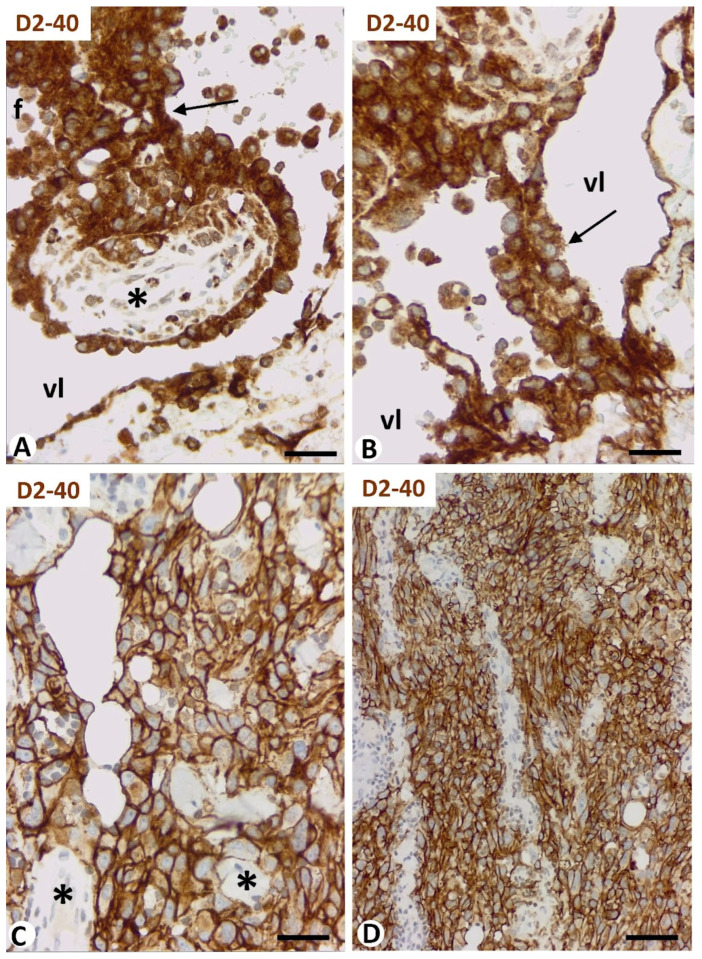
Transitional areas of an angiosarcoma with mixed pattern. Note the progressive increases of the number of layers of neoplastic endothelial cells covering meshes (asterisk, in (**A**)) and pillars (arrow in (**B**)), the neoplasm acquiring a solid pattern formed of the endothelial cells, while the mesh and pillar cores (asterisks, in **C**) gradually become less evident (**C**,**D**). Bar: (**A**–**C**): 10 μm; (**D**): 36 μm.

**Table 1 biomedicines-12-00258-t001:** Common and changing findings of pillars and meshes in vessel tumors. Semiquantitative analysis. Frequency and characteristics of their endothelial cells (endothelial cell morphology -flat/plump-, stratification, mitosis, atypia): 0 (absent, only for endothelial cells characteristics), 1 (<33%), 2 (33–66%), 3 (>66–<100%), and 4 (100%, referring to the tumor type with greater intensity of each estimated fact). Predominant core characteristics: 1: connective tissue, 2: fibrin, 3: complex structures, and 4: inflammatory component. Predominant arrangements: 1: densely grouped, 2: loosely grouped, 3: complex, 4: sinusoidal/linear, and 5: glomeruloid. Predominant mechanisms of formation: 1: vessel loops, 2: endothelialized thrombus, 3: intraluminal endothelial bridges, and 4: splitting and fusion. In lobular capillary hemangioma and Kaposi sarcoma, data only refer to early stages. In angiosarcoma, data correspond to transitional areas between vasoformative and solid areas of the tumor. Asterisk in retiform hemangioendothelioma corresponds to a particular hobnail and matchstick appearance of endothelial cells.

Pillars and Meshes in Blood and Lymphatic Vessel Tumors											
Common findings			Presence of pillars (≤4 µm) and meshes (≥4 µm) Main components of pillar and meshes: a cover formed of ECs and a core formed of connective tissue components. Connections between meshes, meshes and pillars, and between meshes/pillars and the vessel wall								
Variable findings	Benign	Types of tumors	Frequency of pillars and meshes	Characteristics of ECs in pillars and meshes					Core characteristics	Arrangement	Predominant mechanisms of formation
				Flat	Plump	Stratification	Mitosis	Atypia			
		Lobular capillary hemangioma (pyogenic granuloma)	3	3	1	0	2	0	1	1	1
		Intravascular papillary endothelial hyperplasia (Masson tumor)	4	3	1	0	1	0	1, 2	1, 3	1, 2
		Sinusoidal hemangioma	3	4	0	0	0	0	1, 2	2, 4	1, 4
		Cavernous hemangioma	2	3	1	0	0	0	1, 2	2	1, 2
		Glomeruloid hemangioma	3	3	1	0	0	0	1, 2	1, 5	1, 2
		Angiolipoma	2	3	1	0	0	0	1, 2	2	2
		Lymphangioma	3	3	1	0	0	0	1, 3	2	1, 3
	Low-grade	Retiform Hemangioendothelioma	1	0	*	1	1	1	1	2	1, 3
		Dabska tumor	3	0	4	3	2	2	1	1	1, 3
	Malignant	Angiosarcoma	3	0	4	4	4	4	1, 4	1	1, 4
		Kaposi sarcoma	3	2	2	1	1	1	1, 4	2	1

## Data Availability

All data are reported in the present paper.

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
