# Peer review of "Phenomena of Intussusceptive Angiogenesis and Intussusceptive Lymphangiogenesis in Blood and Lymphatic Vessel Tumors"

_biomedicines, 2024, doi:10.3390/biomedicines12020258_

Round 1

Reviewer 1 Report

Comments and Suggestions for Authors

biomedicines-2819242

General comments

In this study, the authors investigated the role of intussusceptive angiogenesis (IA) and intussusceptive lymphangiogenesis (IL) in the growth and morphogenesis of vessels, emphasizing the scarcity of studies on these processes in vessel tumors (VTs). The main goal of this study is to evaluate the presence, characteristics, and potential mechanisms of intussusceptive structures in various benign and malignant blood and lymphatic VTs. The study employs conventional procedures, semithin sections, and immunohistochemistry to examine a diverse spectrum of VTs. Results reveal the presence of intussusceptive structures in benign, low-grade malignancy, and malignant VTs, exhibiting endothelial cover and a core composed of connective tissue components. The origin of these structures is suggested to involve vessel loops, endothelialized thrombus, interendothelial bridges, and/or splitting and fusion. The findings underscore the involvement of IA and IL, alongside sprouting angiogenesis, in the growth and morphogenesis of VTs, providing insights for further molecular studies with therapeutic implications.

Special comments:

1.      The manuscript is primarily descriptive, and the significant deficit lies in the absence of molecular mechanisms within this paper.

2.      The manuscript could be more convincing with the inclusion of quantitative analyses on both histological and fluorescent images.

Author Response

Reviewer 1 

  1. The objective of this paper was the histological demonstration (conventional procedures, semithin sections, and immunochemistry and immunofluorescence microscopy, using different demonstrative markers) of intussusception in a wide range of benign and malignant blood and lymphatic vessel tumors, which has been incompletely and little assessed in the literature. Concerning molecular mechanisms, we explain in the Discussion: “Our contributions in vessel tumors, highlighting the participation of intussusceptive angiogenesis in morphogenesis and vascular network growth, can provide the groundwork for future studies on regulatory molecular mechanisms of intussusception in these diseases and their possible control.”
  2. Following your indications, we have added section 2.4 to Material and Methods, and section 3.5 and Table 1 to Results.

2.4. Semiquantitative analysis

A semiquantitative analysis of the frequency, characteristics of the cover and core, arrangement, and mechanisms of formation of pillars and meshes was performed by two of the authors. The parameters of frequency and characteristics of endothelial cells (endothelial cell morphology -flat/plump-, stratification, mitosis, atypia) were estimated like this: 0 (absent, only for endothelial cells characteristics), 1 (<33%), 2 (33-66%), 3 (>66 - <100%) and 4 (100%, referring to the tumor type with greater intensity of each estimated fact). The predominant core characteristics were: 1: connective tissue, 2: fibrin, 3: complex structures, and 4: inflammatory component. The predominant arrangement of pillars and meshes were: 1: densely grouped, 2: loosely grouped: 3: complex, 4: sinusoidal/linear, and 5: glomeruloid. The predominant mechanisms of formation were 1: vessel loops, 2: endothelialized thrombus, 3 intraluminal endothelial bridges, and 4: splitting and fusion.

3.5. Common and changing events in pillars and cores in benign and malignant blood and lymphatic vessel tumors.

Common and changing events in pillars and meshes were observed in the benign and malignant blood and lymphatic vessel tumor samples included in the present study, although the changing findings could be combined in the same type of tumor. The common findings, especially the presence of these intussusceptive structures in all vessel tumor types, were very important in supporting the involvement of intussusception in vessel tumors. The changing events (frequency, cover and core characteristics, arrangement, and predominant formation mechanisms of intussusceptive structures) varied according to tumor type and were frequently intermingled in the same tumor.  Intravascular papillary endothelial hyperplasia was the vessel tumor (pseudotumor, reactive lesion) with the highest number of pillars and meshes (myriad) and variations in their mechanisms of formation, while transitional areas between vasoformative and solid patterns of angiosarcoma were those with a higher number of mitosis and atypia in the endothelial cells that formed the covers of pillars and meshes. In Table 1 we summarize these findings.

Table 1 is inserted in the manuscript.

Table 1 legend. “Table 1. Common and changing findings of pillars and meshes in vessel tumors. Semiquantitative analysis. Frequency and characteristics of their endothelial cells (endothelial cell morphology -flat/plump-, stratification, mitosis, atypia): 0 (absent, only for endothelial cells characteristics), 1 (<33%), 2 (33–66%), 3 (>66 - <100%) and 4 (100%, referring to the tumor type with greater intensity of each estimated fact). Predominant core characteristics: 1: connective tissue, 2: fibrin, 3: complex structures, and 4: inflammatory component. Predominant arrangements: 1: densely grouped, 2: loosely grouped, 3: complex, 4: sinusoidal/linear, and 5: glomeruloid. Predominant mechanisms of formation: 1: vessel loops, 2: endothelialized thrombus, 3: intraluminal endothelial bridges, and 4: splitting and fusion. In lobular capillary hemangioma and Kaposi sarcoma, dates only are referred to early stages. In angiosarcoma, data correspond to transitional areas between vasoformative and solid areas of the tumor. Asterisk in retiform hemangioendothelioma corresponds to a particular hobnail and matchstick appearance of endothelial cells.”

Thank you for your kind consideration and for your help in improving this work.

Reviewer 2 Report

Comments and Suggestions for Authors

     The article by Díaz-Flores et al. presents an indepth description of intussusceptive phenomena occurring in blood and lymphatic vessel tumors at the level of light microscopy using conventional, immunochemical and immunofluorescent techniques.

     The study is well designed and the methodology properly carried out by means of quite complete histological analysis comprising a wide range of both benign and malignant vessel tumors. The paper has high quality, providing very interesting new data on this topic, which has been very scarcely assessed in the literature. The discussion is very valuable showing a comprehensive and appropriate interpretation of the findings.

     The authors propose dynamic mechanisms to explain the results and for vessel tumor morphogenesis, coming to original conclusions consistent with the findings which are relevant for a better understanding of the pathophysiology of the process. The paper also holds the value of stating the basis for future studies which may contribute to implement measures for the clinical management of cancer.

     Regarding its formal aspects, it is well written and well structured, showing high-quality beautiful photomicrographs which properly illustrate the relevant information. In addition to minor typos which will be corrected when revised by the editorial staff, please make the following corrections:

1) "mesh" instead of "meshe" throughout the text (lines 165, 200, 285, 321, 322, 348 and 396).

2) In figure 1B no arrow is shown, but an asterisk.

3) Line 114: Write "Masson" before "Trichrome".

4) In figure 10E arrow is missing and in its caption "F" must be written instead of "G" (line 366).

5) Lines 461-462: spaces before "connected" and after "vessel".

6) Line 492: "T" (capital) in "the".

Author Response

 Reviewer 2

1)  “meshe” has been replaced by “mesh” in the text and the figure legends

2)  “arrow” has been replaced by “asterisk” in Fig. 1B

3)  Masson has been written before trichrome in Material and methods.

4)  An arrow has been added to Fig 10E and “G” replaced by “F” in the figure legend

5)  Spaces have been inserted before “connection” and after “vessel”

6)  “the” has been replaced by “The”

Thank you for your kind consideration and for your help in improving this work.

Round 2

Reviewer 1 Report

Comments and Suggestions for Authors

Most of my concerns were addressd. it can be accepted for publication.